# Positional End-Point Nystagmus during Positional Testing: Prevalence, Characteristics and Differences with Benign Paroxysmal Positional Vertigo

**DOI:** 10.3390/jcm12010393

**Published:** 2023-01-03

**Authors:** Emilio Domínguez-Durán, Lucia Prieto-Sanchez-de-Puerta, Daniel Iván Martín-Jiménez, Serafin Sanchez-Gomez

**Affiliations:** 1Unidad de Gestión Sanitaria de Otorrinolaringología, Hospital Universitario Virgen Macarena, 41009 Seville, Spain; 2Hospital QuironSalud “Infanta Luisa”, 41010 Seville, Spain

**Keywords:** physiologic nystagmus, benign paroxysmal positional vertigo, Dix–Hallpike test, positional maneuvers

## Abstract

Background: Some individuals present positional end-point nystagmus when the Dix–Hallpike tests are performed on them if they unintentionally look towards the examined ear. Objective: To describe the prevalence and the characteristics of end-point nystagmus during positional testing in healthy subjects. Methods: Sixty healthy subjects were included. Eight positional tests were performed on them, two Pagnini–McClure tests and six Dix–Hallpike tests, while keeping the eyes in different positions; one on each side. Two independent observers filled in a questionnaire about the presence of positional nystagmus, its latency, duration, direction, and sense. Results and conclusions: Of the subjects, 65% showed positional end-point nystagmus. This nystagmus had a short latency and last for as long as the head is maintained in the test position. They can show any direction or sense, but the most common are torsional clockwise in left tests and anticlockwise in right tests. Unlike BPPV, this nystagmus did not appear with the eyes in the straight-ahead position, it is asymptomatic, and its intensity does not decline.

## 1. Introduction

The inclusion of benign paroxysmal positional vertigo (BPPV) in the differential diagnosis of episodic vestibular syndrome has transformed the pre-existing protocols in clinical practice over the last few decades. Nowadays, BPPV is considered the most common cause of vertigo [1,2]. BPPV is diagnosed by having a classical case history compatible with BPPV and by observing canal-specific positional nystagmus during head positional maneuvers [3], such as those described by Pagnini, McClure or Dix and Hallpike [4,5,6].

In our daily practice, positional tests are performed on most of the patients that have a neurotological consultation, due to their simplicity, immediacy, and the high chance of diagnosing and treating BPPV. However, sometimes a positional nystagmus may appear during testing that is difficult to interpret. The observed positional nystagmus has to meet specific predefined criteria for it to be assumed that it is caused by BPPV [3]. Differences in positional nystagmus are often noticed among patients, and these can be explained by the different possible starting positions of the otoconia, the geometry of the semicircular canals or the density and radii of the dislodged fragments [7]. Despite this variability, not all cases of positional nystagmus can be explained by BPPV. Other less known entities can also cause positional nystagmus, such as migraine, injury near the cerebellar nodulus, vertebral artery compression, pressure-induced inner ear fluid disorders or labyrinthine hypofunction [8].

With these differential diagnoses, it has not yet been established whether positional nystagmus without pathological significance may appear in healthy subjects. Our previous observations showed that some individuals may present torsional end-point nystagmus when BPPV testing is performed on them and they unintentionally look towards the examined ear; i.e., when they fix their gaze on the floor. End-point nystagmus is defined as a physiologic gaze-evoked nystagmus in the absence of pathology, attributed to normal variation in gaze-holding ability. It is typically seen with extreme lateral gaze and is generally low amplitude, low frequency, binocular, symmetric in right and left gaze, poorly sustained and unassociated with other ocular motor or neurological abnormalities [9]. It has been defined with the patient upright.

The objective of this study was to determine the prevalence of, and describe the characteristics of, the end-point nystagmus during positional testing in healthy subjects who have never suffered from vertigo or other vestibular symptoms, as well as, to compare its characteristics with those of BPPV, in order to provide a proper differentiation and avoid wrong diagnoses.

## 2. Materials and Methods

This was a prospective observational study performed with healthy volunteer subjects.

Firstly, the sample size was calculated. Our previous non-systematized observations had found that a physiologic end-point nystagmus during the Dix–Hallpike testing appeared in about 50% of the population. We expected to obtain a confidence interval of the population who showed these types of positional nystagmus patterns with a range of 25 percentual points. Then, the sample size required was estimated to be 60 subjects.

Secondly, the staff of our medical centers, as well as their patients and companions, were asked to participate in the study. Volunteers were chosen based on gender and age criteria. Ten groups of six subjects were selected, based on their gender (male or female) and their age by decades (from 20 to 29 years of age, and then increasing by 10 years in each group up to the group including those over the age of 60). We finally defined five groups of different age ranges (10 years each) with six males and six females. This sampling technique guaranteed a uniform distribution of patients in these categories.

Then, all the volunteer subjects were screened using a systematized checklist that included all the inclusion and exclusion criteria. These criteria are listed in Table 1. Those subjects who did not meet all of the inclusion criteria, or those who met any of the exclusion criteria, were not included or excluded from the study.

Following inclusion, a video head impulse test (vHIT) was performed on all of the remaining volunteers. These tests were done by the same tester (an experienced otolaryngologist) according to the usual protocol of our centers, in which the patient is seated in a solid chair, one meter away from a fixed visual stimulus on a plain wall and with the same lightning. Then, the device is calibrated and finally 10 head impulses in each sense of the yaw plane are performed, while the patient is asked to keep their gaze fixed on the stimulus. These impulses were performed fast, abruptly, unpredictably and with a maximum angular velocity between 200 °/s and 300 °/s. This test allowed us to calculate the gain of the vestibulo–ocular reflex of the left and right horizontal semicircular canals. If the values of any of these gains were below 0.80 or above 1.20, values considered normal according to the specifications of the device, the subject was excluded from the study.

Then, eight positional tests were performed on each volunteer. The transition time of movement from one position to another was approximately 5 s. One Pagnini–McClure test [4,5] was performed on each side, while keeping the eyes in straight-ahead position, one Dix–Hallpike test [6] was performed on each side, asking the subject to keep the eyes in the straight-ahead position from the beginning of the test (hereafter referred to as SAP tests), one Dix–Hallpike test was performed on each side, asking the subject to keep the eyes on the horizontal end-point of the side of their examined ear from the beginning of the test (hereafter referred to as HEP tests) and one Dix–Hallpike test was performed on each side. asking the subject first to keep the eyes in the straight-ahead position and then, when the head was in the final position, to turn the eyes on to the horizontal end-point position of the side of their examined ear (hereafter referred to as SAP to HEP tests). These tests are illustrated on Figure 1. The subjects were asked to keep their eyes open and to blink as little as possible. The left tests were performed first on the odd-numbered subjects of each group, whereas the right tests were performed first on the even-numbered subjects to determine whether there was a possible effect of the fatigability of the positional nystagmus on the results. A flowchart of the protocol of the study is shown on Figure 2.

All of the positional tests were performed using video glasses (ICS Impulse^®^, Optomic, Colmenar Viejo, Spain that allowed for a more detailed and magnified examination of the eye. Recordings were taken from the right eye. These video glasses did not prevent the fixation of the gaze. We asked the subjects to maintain the gaze in the straight-ahead position without looking to any lockable target. In the case of looking to the floor, it was a uniform monochrome pattern on which the subject was not able to fix his or her. This was an intentional part of the design of the study to try to ensure that the conditions of the experiment resembled the conditions we use when the BPPV test is performed on dizzy patients. The positions during the tests were maintained for 40 s [3].

During the positional tests, three clinicians were present in the examination room. One of them was the one in charge of doing all of the positional tests and this person was asked not to communicate with the other two clinicians. The other two clinicians were experienced neurotologists who acted as observers and who had to fill out a standardized questionnaire, in which they were asked whether they observed any nystagmus in each of the positional tests, as well as its latency, duration, direction and sense. They were asked to fill out their own questionnaires independently of each other to determine the inter-rater reliability of their observations. If any nystagmus was observed during positional testing, then the patient was asked if they experienced any concomitant vertigo. In the event of an incidental unintentional diagnosis of BPPV by any of the three clinicians, due to the presence of symptoms and nystagmus compatible with any of the variants of BPPV described by the Bárány Society in their criteria [3], the patient was excluded from the study.

Finally, the prevalence and characteristics of end-point nystagmus during positional testing in this sample were calculated using the software SPPS for Mac. The nystagmus was described using the recommendations given by the Bárány Society [9]. The prevalence of positional end-point nystagmus was then analyzed, whilst taking into account the following characteristics of the subjects: gender, age, gain of the vestibulo–ocular reflex, presence of saccades in the vHIT, presence of positional nystagmus during the SAP tests, position of the eyes during the movement of the tests and the side that was first examined. The degree of concordance between the two observers in their ratings was also analyzed.

## 3. Results

A total of 137 volunteers were recruited in order to have at least 60 patients in the final sample. Four volunteers were excluded because their age and gender group already had six volunteers and 48 volunteers were excluded because they had suffered from some kind of vertigo in the past or they were suffering from it at the time of inclusion. Six volunteers were excluded because they felt dizzy on the day of testing. One volunteer had had ear surgery. Four volunteers were excluded because they were under the influence of drugs that alter vestibular function. Fourteen volunteers were excluded because abnormal values of the gain of the vestibulo–ocular reflex were found during the video head impulse test in one of their ears. The algorithm used to select the final sample can be seen in Figure 3.

With the description of the observed positional nystagmus, the eyes were used as a frame of reference. When describing torsional nystagmus, the terms clockwise and anticlockwise are used as seen from the clinician’s perspective.

### 3.1. Prevalence of Positional Nystagmus and Inter-Rater Concordance

First, the Pagnini–McClure tests were performed on all of the patients on the left and right sides. None of the raters observed any positional nystagmus in any of the 120 tests and the percentage of concordance was 100%.

Then, SAP tests were performed and the raters agreed in 98.5% of cases (Cohen’s *κ* coefficient = 0.49). In 117 of the 120 tests, none of the raters observed any positional nystagmus. In one case, both raters observed a vertical downbeat positional nystagmus, without associated symptoms, without latency and that lasted for more than 40 s, and which did not fulfill the criteria for BPPV. The raters disagreed in the following two cases: rater #1 observed one case of horizontal geotropic positional nystagmus and one case of horizontal apogeotropic positional nystagmus which were not observed by rater #2.

Later, HEP tests were performed and the raters agreed about 92.5% of them (Cohen’s *κ* coefficient = 0.85). During the left tests, the raters agreed about the presence of positional nystagmus in 28 cases (46.7% of the sample; 95% CI (32.3–58.3%)). They agreed on their direction in all of these cases and about their sense in 96.4% of them (in one case, one rater observed a torsional clockwise positional nystagmus that was described as anticlockwise by the other rater). If one only looked at the 27 cases where both raters agreed, 70.4% of these cases presented a torsional clockwise positional nystagmus. The other kinds of observed positional nystagmus are shown in Table 2. The latency and the duration of positional nystagmus was not normally distributed and there was no significant difference between the observations of the raters regarding these variables (Wilcoxon signed-rank test *p* = 0.15 and *p* = 0.09, respectively). The median latency of the positional nystagmus was two seconds and its median duration was 38 s. There were three cases where the raters disagreed about the presence of positional nystagmus. There was one case in which only rater #1 observed a clockwise positional nystagmus, one case in which only rater #2 observed an anticlockwise positional nystagmus and one case in which rater #1 observed a downbeat positional nystagmus. During the right tests, the raters agreed about the presence of positional nystagmus in 29 cases (48.3% of the sample; 95% CI (35.4–61.4%)). They agreed about the direction and sense in 23 of these cases (79.3%), but there were five cases where they disagreed about the direction (two cases where there was debate as to whether it was horizontal or torsional positional nystagmus and three cases where they debated between vertical or torsional positional nystagmus) and one case in which they disagreed about the sense of the torsional positional nystagmus. If one only looks at the 23 cases where they agreed, 47.8% of these cases presented a torsional anticlockwise positional nystagmus. The other cases are shown in Table 2. The latency and distribution of positional nystagmus was not normally distributed. There was no significant difference between the observations of the raters regarding these variables (Wilcoxon signed-rank test *p* = 0.409 and *p* = 0.171, respectively). The median latency of the positional nystagmus was three seconds and its median duration was 37 s. There were six cases where the raters did not agree about the presence of positional nystagmus. Only rater #1 observed two cases of torsional anticlockwise positional nystagmus, one case of horizontal geotropic positional nystagmus and one case of vertical downbeat positional nystagmus, while only rater #2 observed one case of torsional clockwise positional nystagmus and another case of torsional anticlockwise positional nystagmus.

Finally, SAP to HEP tests were performed. The raters agreed in 87.5% of them (Cohen’s *κ* coefficient = 0.75). During the left tests, the raters agreed about the presence of positional nystagmus in 29 cases (48.3% of the sample; 95% CI (35.4–61.5%)). They agreed about their direction in 24 (82.8%) of the cases. In four cases, the raters disagreed about the direction of the positional nystagmus, and they observed horizontal, torsional, and vertical positional nystagmus. In one case, they disagreed about the sense of the observed torsional positional nystagmus. If one only looks at the 24 cases where both raters agreed, 70.8% of these cases presented a torsional clockwise positional nystagmus. The rest of the cases are in Table 2. The latency and distribution of positional nystagmus was not normally distributed and there was no significant difference between the observations of the raters regarding these variables (Wilcoxon signed-rank test *p* = 0.969 and *p* = 0.379, respectively). The median latency of the positional nystagmus was 1.5 s, and its median duration was 38 s. There were six cases where the raters disagreed about the presence of positional nystagmus. There were two cases in which only rater #1 observed a vertical upbeat positional nystagmus and a downbeat positional nystagmus, respectively, three cases in which only rater #2 observed a torsional clockwise positional nystagmus and one case in which only rater #2 observed a vertical downbeat positional nystagmus. During the right tests, the raters agreed about the presence of positional nystagmus in 30 cases (50.0% of the sample; 95% CI (36.9–63.1%)). They agreed about the direction and sense in 25 of these cases (83.3), but there were five cases in which rater #1 observed horizontal geotropic, apogeotropic, downbeat or clockwise positional nystagmus while rater #2 observed anticlockwise positional nystagmus in all of these cases. If one only looks at the 25 cases where they agreed, 60.0% of these cases presented a torsional anticlockwise positional nystagmus. The prevalence of the other directions can be seen in Table 2. The latency and distribution of positional nystagmus was not normally distributed and there was no significant difference between the observations of the raters regarding these variables (Wilcoxon signed-rank test *p* = 0.243 and *p* = 0.133, respectively). The median latency of the positional nystagmus was one second and its median duration was 39 s. There were nine cases in which only one of the raters observed a positional nystagmus. Rater #1 observed two cases of horizontal geotropic positional nystagmus, one case of horizontal apogeotropic positional nystagmus and one case of torsional clockwise positional nystagmus, while rater #2 observed three cases of torsional anticlockwise positional nystagmus, one case of torsional clockwise positional nystagmus and one case of vertical downbeat positional nystagmus.

The prevalence of positional end-point nystagmus was calculated. These positional nystagmus cases were detected by the two raters in at least one of the tests in 65.0% of the population sample (95% CI (51.2–76.6%)). When a subject presented positional nystagmus during any of the tests, they were asked whether they were experiencing vertigo. All of the cases were asymptomatic for the duration of all of their positional nystagmus and no incidental cases of BPPV were detected.

Some examples of the positional nystagmus observed during the study are shown in the Appendix A. Appendix A shows a torsional clockwise positional nystagmus and a torsional anticlockwise positional nystagmus and Appendix A shows other less frequently observed cases of positional nystagmus.

### 3.2. The Persistence of Positional End-Point Nystagmus during the Different Maneuvers

We analyzed how the presence of nystagmus in any of the positional tests could influence the presence of positional nystagmus in one of the others. We selected the tests in which positional end-point nystagmus were detected, i.e., HEP tests and SAP to HEP tests. We considered that positional nystagmus was present only in those cases where both raters agreed about its presence, direction, and sense. We considered that positional nystagmus was not present in the rest of the cases. Figure 4 shows how the subjects were distributed when both raters agreed in the positional tests. Table 3 shows the degree of concordance in the different tests measured using the percentage of cases they both agreed upon and Cohen’s *κ* coefficient. The patient who presented a vertical downbeat positional nystagmus during the standard right Dix–Hallpike test showed the same positional nystagmus in the left HEP test and a horizontal geotropic positional nystagmus in the left SAP to HEP test but did not present any positional nystagmus in the right tests.

### 3.3. Influence of Age and Gender on the Presence of Positional End-Point Nystagmus

The relationship between the age and the presence of positional end-point nystagmus in any of the tests was first analyzed using an univariant binary logistic regression model. This model gave an e^*β*^ value of 0.986 with a 95% CI of (0.950–1.023). In light of these results, any relationship between age and positional end-point nystagmus was discarded.

Then, the relationship between gender and the presence of positional end-point nystagmus was calculated using the Fisher’s exact test. No significant difference was found between the percentage of males with positional end-point nystagmus (56.7%) and the percentage of females (73.3%) and a *p*-value of 0.279 was obtained. The relationship between the number of positive tests and the gender was also calculated, but this relationship did not reach statistical significance either, with a *p*-value of 0.314 obtained using the Mann–Whitney *U* test.

### 3.4. The Relationship between Positional End-Point Nystagmus and the Characteristics of the vHIT

The variables and the relationship between them next studied were the presence of positional end-point nystagmus detected by both raters and the characteristics of the vHIT.

Firstly, the values of the gain of the vestibulo–ocular reflex were studied. Using a binary logistic regression model, the relationship between the gain of the left horizontal semicircular canal and the presence of positional end-point nystagmus when performing HEP or SAP to HEP tests did not reach statistical significance (e^*β*^ values of 1.006 and 1.008 with a 95% CI of (0.950–1.066) and (0.950–1.068), respectively). The relationship between the gain of the right horizontal semicircular canal and the presence of this nystagmus was not statistically significant either (e^*β*^ values of 0.998 and 1.046 with a 95% CI of (0.938–1.062) and (0.981–1.115), respectively).

Next, the presence of saccades in the vHIT was classified using the following three levels: absence of saccades, presence of saccades whose height was less than half of the size of the peak head velocity, and presence of saccades whose height was over half of the side of the peak head velocity. A binary logistic regression model was used to test these relationships and the relationship between the presence of saccades when testing the left horizontal semicircular canal and the presence of positional end-point nystagmus when performing HEP or SAP to HEP test was not statistically significant (e^*β*^ values of 0.445 and 0.660 with a 95% CI of (0.179–1.107) and (0.276–1.577), respectively). There was no statistically significant relationship between the presence of saccades when testing the right horizontal semicircular canal and positional end-point either (e^*β*^ values of 0.789 and 1.281 with a 95% CI of (0.351–1.772) and (0.595–2.757), respectively).

### 3.5. The Influence of the First Side Tested on the Presence of Positional End-Point Nystagmus

Finally, the relationship between the first side tested and the presence of positional end-point nystagmus was analyzed. No significant differences in the prevalence of the different nystagmus observed by the two raters were found to be related to the first side tested. When the left side was the first to be tested, the prevalence of positional end-point nystagmus in the left and right HEP and SAP to HEP tests were 43.3%, 26.7%, 36.7% and 33.3%, respectively. When the right side was the first to be tested, these percentages were 46.7%, 50.0%, 43.3% and 50.0%. None of the *p*-values of the Fisher’s exact test were statistically significant.

## 4. Discussion

### 4.1. Characteristics of the Positional End-Point Nystagmus

The main conclusion of this study that the authors wish to emphasize is that in 65% of the population, that has not suffered from vertigo and has a normal vestibular function, positional end-point nystagmus is noticeable enough to be detected by two raters. Previous studies found that up to 88% of the healthy population, not showing nystagmus when the gaze is fixed, might have positional nystagmus in the neutral position of the gaze when fixation was not possible [10]. Our study, however, intended to detect other kinds of nystagmus, not purely related to the position, but only appearing in the combination of position of the head and the extreme gaze. This nystagmus can even be observed without using devices that prevent the fixation of the gaze. Therefore, we believe that these percentages are not comparable.

This nystagmus has a latency that normally lasts from one to three seconds and, in most cases, it lasts for as long as the head is kept in the test position. This nystagmus can show any direction or sense, but torsional nystagmus is the most common, beating clockwise in the left tests and anticlockwise in the right ones. The second most frequently observed nystagmus was vertical downbeat. Due to its direction and sense, it can easily be mistaken for the nystagmus caused by BPPV but, unlike that of BPPV, it does not appear when the test is performed with the eyes in the straight-ahead position, it is always asymptomatic, and its intensity does not seem to decline during the test.

This nystagmus does not seem to show any relationship to any of the examined variables. We found no statistically significant relationship between the appearance of the positional end-point nystagmus according to gender, age, the characteristics of the vHIT or the first side tested.

The occurrence of a positional end-point nystagmus in any of the tests did not necessarily mean it would occur in any other tests, but, according to the *κ* index, there was a moderate association between its appearance in a HEP test and an SAP to HEP test on the same side. The association in the appearance of this nystagmus was also moderate if we look at the same test (HEP or SAP to HEP) on different sides. The association was between fair and moderate when neither the side nor the type of test performed were the same [11].

### 4.2. Subjective Aspects Related to the Detection of the Positional Nystagmus in the End-Point Position

The previously described positional end-point nystagmus was seen by only one rater in between 3.3% and 15.0% of the cases in which the tests were performed. There are two possible reasons for this. Firstly, this nystagmus is sometimes subtle because it may have low amplitude. Secondly, determining the direction and sense of the eyes when they are in the end-point position may be difficult because when the eyes are in this position, the anatomical references of the sclera are partially hidden and the projection of the circle pupil in the front plane is elliptical.

One might argue that this nystagmus could have been detected using videonystagmography; however, automated systems have not been programmed to measure nystagmus in these positions and the partial detection of the pupil would affect the results obtained. Videonystagmography may sometimes be less reliable than a direct examination of the eyes, especially when dealing with torsional nystagmus, as the torsional component of the nystagmus can be hard to interpret [12,13]. These factors influenced our decision to design a study where nystagmus was evaluated by two independent raters, and, in this case, there was a moderate degree of inter-rater concordance.

### 4.3. The Proposed Physiopathological Mechanism Causing Positional End-Point Nystagmus

The Bárány Society’s current classification of nystagmus describes end-point nystagmus as a physiologic nystagmus that occurs in the absence of pathology and is attributed to a normal variation in gaze-holding ability. It is described as a low-amplitude, low-frequency, and poorly sustained nystagmus that may occasionally be sustained [9].

Our first hypothesis was that the inner ear must have a role in the development of positional end-point nystagmus because it senses the position of the head. Thus, the left Dix–Hallpike position produced an excitation of the left posterior semicircular canal that produces a torsional clockwise nystagmus. On the other hand, the right Dix–Hallpike position activated the right posterior canal and generates a torsional anticlockwise nystagmus. This theory was supported by our findings that the presence of positional end-point nystagmus was not related to the position of the eye during the maneuver. However, this hypothesis did not explain the sustainability of the nystagmus, whose intensity should have declined over time, and it did not explain the presence of other directions or senses of nystagmus when present either. This counterevidence to the hypothesis could have been explained by a potential imbalance of vestibular function or a cerebellar dysfunction in the regulation of the brainstem [14], but, arguably, these explanations may be discarded as we only studied subjects without neurotological symptoms at the time of testing, without antecedents of neurotological diseases, and who had a normal vHIT. The presence of positional end-point nystagmus did not increase with age, making the idea that a hypothetical brainstem dysfunction could have caused it seem less plausible.

As it seems improbable that the inner ear is the main cause of positional end-point nystagmus, we preferred to explain this phenomenon as being a consequence of the function of the neurons responsible for the tonic innervation of ocular muscles. Whilst the eye position is kept on the end-point, the ocular muscles must control the elastic restoring forces of the orbit. The function of the neurons that innervate these muscles is the result of the output of the neural integrator, which is responsible for the integration of the primary vestibular afferents and the eye position in the orbit [15]. At the same time, the occurrence of end-point nystagmus is caused by the velocity of the slow phase ocular drift during fixation to stabilize the gaze [16]. Therefore, the occurrence of positional end-point nystagmus would depend on both the velocity of the slow phase ocular drift and the function of the central neural integrator, depending on the vestibular input and the position of the eye.

### 4.4. Strengths and Limitations of the Study

The main strength of this study was its careful and systematic method. The study obtained a protocolized and systematic exploration of all the patients who participated in it. We developed our work in order to elucidate the origin of the positional end-point nystagmus by designing tests to separate the roles of the head movement, the ocular position during the test, the ocular movement, and the vestibular function. The most difficult objective variables were evaluated by independent observers, and, thus, the results increase their reliability and extrapolation to other clinicians.

However, after having analyzed the results, new questions arose that should be resolved in further studies. It remains unknown how vestibular function affects this nystagmus, its characteristics in patients with altered oculomotor function and how it interferes with other pathologies that cause positional nystagmus.

### 4.5. Application of the Results in Clinical Practice

After describing that 65% of the population may have any kind of positional end-point nystagmus, it is important to know that it is common in the population and that it is different from that caused by BPPV. From our point of view, it is crucial to differentiate both types of nystagmus to avoid incorrect diagnoses caused by the increased use of the BPPV diagnostic tests. The results obtained prompted us recommend that it is necessary to keep the eyes in neutral gaze position during BPPV testing, because if they are in the end-point position it is possible to find some positional nystagmus without pathological significance that can cause a false diagnosis of BPPV. However, it should be kept in mind that there are other types of pathological nystagmus that only appear in the extreme gaze. Therefore, the appearance of these positional nystagmus in the Dix–Hallpike test should not be considered as an innocent finding in all the cases.

## 5. Conclusions

In the healthy population 65% showed positional end-point nystagmus, regardless of their age, gender, order in which the positional tests were performed and characteristics of the vHIT. The positional end-point nystagmus observed had a short latency and lasted for as long as the head was kept in the test position. They could have any direction or sense; the most common ones being torsional clockwise in the left tests and anticlockwise in the right ones. Unlike BPPV, this nystagmus did not appear with the eyes in the straight-ahead position, patients were asymptomatic, and nystagmus intensity did not decline over time.

## Figures and Tables

**Figure 1 jcm-12-00393-f001:**
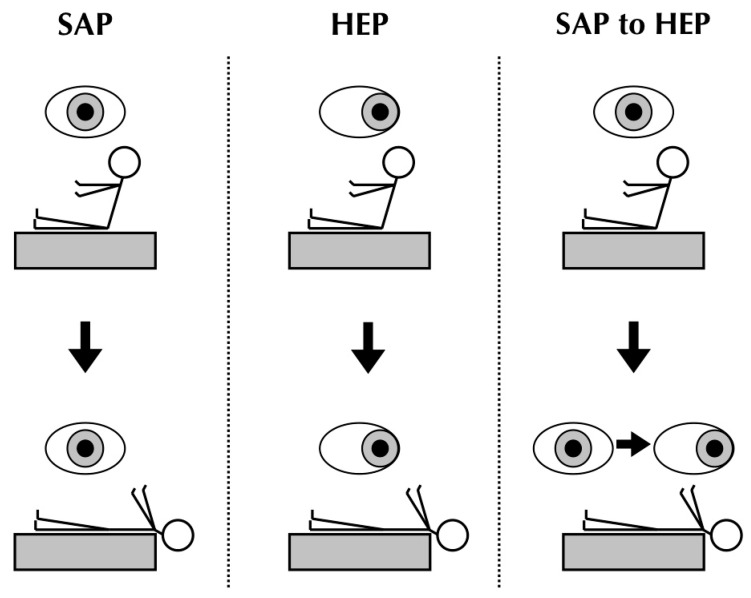
Positional tests performed. The figure represents the different positions of the head and the gaze of each subject during the tests.

**Figure 2 jcm-12-00393-f002:**
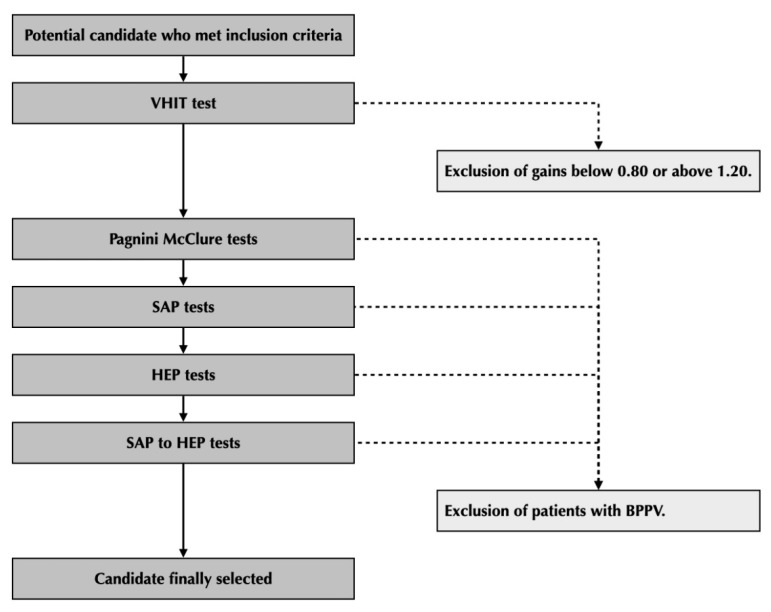
Flowchart of our protocol. The different inclusion and exclusion steps and the performed tests are summarized.

**Figure 3 jcm-12-00393-f003:**
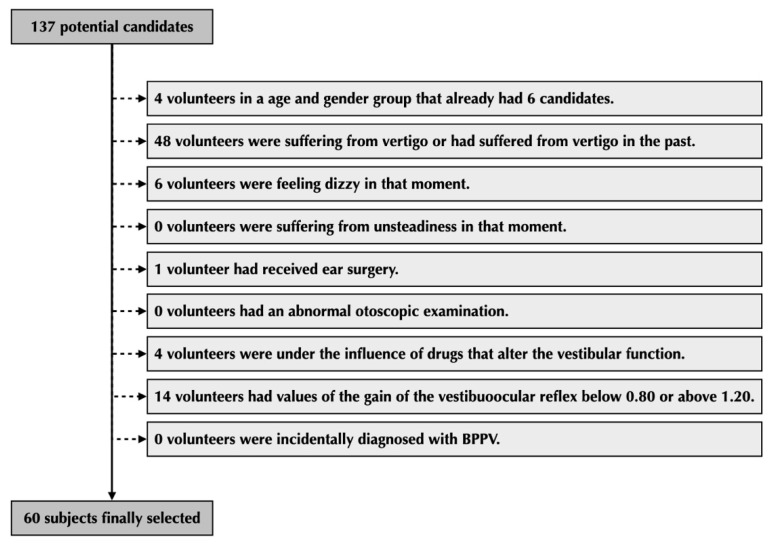
Algorithm used to select the 60 patients that were part of the final sample.

**Figure 4 jcm-12-00393-f004:**
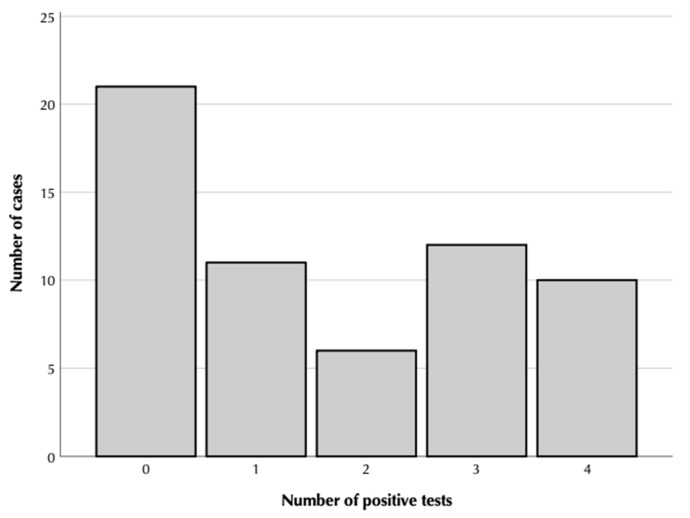
Distribution of cases according to the number of positive tests with the eyes in the end-point position in any part of them.

**Table 1 jcm-12-00393-t001:** Inclusion and exclusion criteria for participation in this study.

Inclusion Criteria	
Age 20 or above.	
**Exclusion Criteria**	
Sufficient amounts of candidates already included (age/gender).	The subject suffers from vertigo or has suffered from it.
The subject feels dizziness during the test.	The subject suffers from some kind of unsteadiness.
The subject has had previous ear surgery.	The otoscopic examination finds other morphological tympanic membrane alterations besides a monomeric tympanic membrane and/or myringosclerosis.
The patient is under the prescription medication of drugs that alter vestibular function (e.g., vestibular suppressants and benzodiazepines).	Pathological lateral semicircular canal Video Head Impulse Test VOR mean gain values (bellow 0.80 or above 1.20).
Fulfilment of BPPV diagnostic criteria.	

**Table 2 jcm-12-00393-t002:** Main results of this study.

	McClure	SAP Tests		HEP Tests		SAP to HEP Tests	
Percentage of concordance and *κ* value	100%	98.5% *κ* = 0.49		92.5% *κ* = 0.85		87.5% *κ* = 0.75	
Examined ear	Any	Left	Right	Left	Right	Left	Right
None of the raters saw any positional nystagmus	60	60	57	29	25	25	21
Only one of the raters saw a positional nystagmus	0	0	2	3	6	6	9
Both raters saw a positional nystagmus, but they disagreed about its direction or sense	0	0	0	1	6	5	5
Both raters saw a positional nystagmus and they agreed about its direction and sense	0	0	1	27	23	24	25
• Torsional anticlockwise	0	0	0	1	11	2	15
• Torsional clockwise,	0	0	0	19	2	17	1
• Horizontal geotropic	0	0	0	1	1	2	2
• Horizontal apogeotropic	0	0	0	1	0	1	1
• Vertical upbeat	0	0	0	0	0	1	1
• Vertical downbeat	0	0	1	5	9	1	5
Median latency	-	-	0 s	2 s	3 s	1.5 s	1 s
Median duration	-	-	40 s	38 s	37 s	38 s	39 s

**Table 3 jcm-12-00393-t003:** Concordance of positional end-point nystagmus. Percentage of concordance and Cohen’s *κ* coefficient in the tests in which positional end-point nystagmus was detected. The direction and sense of the nystagmus were not considered when calculating the degree of inter-rater concordance.

	Left HEP Test	Right HEP Test	Left SAP to HEP Test
Right HEP test	73.3% *κ* = 0.454		
Left SAP to HEP test	78.3% *κ* = 0.558	75.0% *κ* = 0.476	
Right SAP to HEP test	66.6% *κ* = 0.322	80% *κ* = 0.584	71,6% *κ* = 0.414

## Data Availability

Not applicable.

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
