# Peer review of "Positional End-Point Nystagmus during Positional Testing: Prevalence, Characteristics and Differences with Benign Paroxysmal Positional Vertigo"

_jcm, 2023, doi:10.3390/jcm12010393_

Round 1

Reviewer 1 Report

This study explored the incidence and characteristics of the end-point nystagmus evoked by the positional tests in normal subjects with a rigorous design and some research implications. Overall I agree with the main points revealed by the current study, but there are still some issues that prevent the article from being published in its current form.

1.The inclusion and exclusion criteria for normal subjects could be more stringent, e.g. 1) the authors didnt take migraine into account. As we know,  a variety of patterns of nystagmus can be seen in patients with migraine even without current attacks. 2) Normal vHIT does not fully represent normal vestibular function. The article did not explain why "under 80 % or over 120 %" was excluded. vHIT is influenced by the testers' practice, was this study performed by the same testers?

2. In the study methods part, patients underwent 8 positional tests without describing how long each interval was? Does the subjective fatigue of the subject's positional tests affect the results of the experiment? The direction of the patient's gaze during the pagnini-McClure test was not described. Did the investigators observe the presence of nystagmus during extreme eye deviation to the non-test ear? Are the results in Table 2 based on the average number of end-point nystagmus per subject over 8 positional trials? It would be better if a reader-friendly flowchart of the study protocol could be made.

3.Some possible confounding factors are to be elucidated. Did the subjects gaze at the target during the tests? The number of degrees of eye deflection per subject? This paper only focused on the incidence and nystagmus direction of the terminal nystagmus, based on the video of a typical terminal nystagmus of the study provided by the authors, not knowing which instrument the authors used to record it. Actually there exists instruments that are able to record the 3D characteristics of nystagmus, and we would also be interested with the difference of amplitudes and frequencies of positional terminal nystagmus between different vestibular disorders.

4.  I am not quite understand why the Fishers exact test was used for examining the relationship between gender and the presence of end-point nystagmusand the relationship between the first side tested and the presence of nystagmus. (Line 237 and 268).

5. Please describe SAP tests, HEP tests, and SAP to HEP tests in more details. It would be better if these experiments can be illustrated with figures.

6. Sampling from staff, patients and patients’ companions in one medical center did not completely fit the healthy population that the study wants to investigate. And there might be a selection bias.

7. There was a lack of table, figure or literal description to present the demographic features of overall individuals enrolled.

8. There might be a more detailed description of the stratified sampling technique to confirm it was random.

9. Advantages and drawbacks of the proposed approach should be highlighted.

Minor problem:

1. the description of Figure 2 should be more detailed in the manuscript, and whether there is a statistical differences please show it in the figure.

2. Figure 1, typo error, above 80%.

3. No certain information on location and date was mentioned in the part of materials and methods.

Author Response

Dear Dr./Mr./Ms.,

Thank you for giving us the opportunity to submit a revised draft of our manuscript. We appreciate the time and effort that you and the other reviewers have dedicated to providing your valuable feedback on the manuscript. We are grateful to the reviewers for their insightful comments on our paper. We have been able to incorporate changes to reflect most of the suggestions provided by the reviewers.

Here is a point-by-point response to your comments and concerns.

1.The inclusion and exclusion criteria for normal subjects could be more stringent, e.g. 1) the authors didn’t take migraine into account. As we know, a variety of patterns of nystagmus can be seen in patients with migraine even without current attacks. 2) Normal vHIT does not fully represent normal vestibular function. The article did not explain why "under 80 % or over 120 %" was excluded. vHIT is influenced by the testers' practice, was this study performed by the same testers?

Thank you very much for your comment.

In first place, it is true that the presence of migraine was not taken into account in this study and, also, that some subjects with migraine may have positional nystagmus. However, we believe that this does not invalidate the results of our study. The aim of our experiment is to study how the end-point nystagmus looks like in subjects with normal vestibular function, regardless of the presence of migraine. We think that studying the differences of these nystagmus between subjects with migraine and those without migraine can be potentially interesting, but this was not the aim of our study. However, this consideration will be taking into account in our following studies.

Second, it is true that the normal gain on the VHIT test is not sufficient to ensure that a vestibular function is normal. However, we believe that the combination of a normal gain in a VHIT test, in a subject who has never suffered from neurotological pathology, who has never experienced vertigo and who does not present dizziness nor instability and who does not take medication that can suppress vestibular function is enough to assume normal vestibular function. The gain values 0.80 and 1.20 were selected because these were the normal gain values that device specifications recommended to be considered as the normal ones.

Finally, we want to clarify that the VHIT tests carried out were carried out by the same tester, who was the one who also carried out all the BPPV tests, who is an experienced clinician.

  1. In the study methods part, patients underwent 8 positional tests without describing how long each interval was? Does the subjective fatigue of the subject's positional tests affect the results of the experiment? The direction of the patient's gaze during the pagnini-McClure test was not described. Did the investigators observe the presence of nystagmus during extreme eye deviation to the non-test ear? Are the results in Table 2 based on the average number of end-point nystagmus per subject over 8 positional trials? It would be better if a reader-friendly flowchart of the study protocol could be made.

Thank you for your appreciation, we believe it is important to clarify the methodology as much as possible.

  • Regarding the duration of each interval, each tests were performed during 40 seconds. This information is provided in line 97.
  • We considered the effect of the fatigue in the design of the experiment; this was one of the reasons why some individuals were tested first their left side and other their right side. Since the first tested side did not significantly affect the incidence of nystagmus, we concluded that the effect of fatigue is not relevant enough and that it can be ignored without affecting the results of the study.
  • We have realized that we did not describe the direction of the patient’s gaze during the Pagnini-McClure test; in these tests, the eyes were kept in the straight-ahead position. This description has been added to the manuscript.
  • We did not evaluate the presence of nystagmus during extreme eye deviation to the non-test ear and this was based on our previous clinical experience. Before the design of the study, we had noticed that patients that attended to our clinic tend to look to the floor while they were being tested. This probably occurred because when someone is being pushed though some direction, they involuntary tend to look towards it. It would have been interesting to have recorded the behaviour of the eye in the suggested position; however, this was not performed.
  • The results in table 2 are based on the number and percentage of subjects which showed nystagmus; the legend of the table has been changed to notice this.
  • As suggested, a flowchart of the study protocol has been added as Figure 1.

3.Some possible confounding factors are to be elucidated. Did the subjects gaze at the target during the tests? The number of degrees of eye deflection per subject? This paper only focused on the incidence and nystagmus direction of the terminal nystagmus, based on the video of a typical terminal nystagmus of the study provided by the authors, not knowing which instrument the authors used to record it. Actually there exists instruments that are able to record the 3D characteristics of nystagmus, and we would also be interested with the difference of amplitudes and frequencies of positional terminal nystagmus between different vestibular disorders.

Thank you very much for these considerations. We will discuss them separately:

  • During the SAP tests, the subjects were asked to keep the gaze in the straight-ahead position; this was done looking at our plain wall, without any lockable target. During the HEP tests, the subjects were asked to look toward the horizontal end-point of the side of the examined ear; this mostly means towards the floor. Our floor has a uniform monochromed pattern in which the patient is not able to fix its gaze. In case of deviation of the gaze thorough the test, the subject is asked to keep its gaze in the requested position. This acclaration has been added to the manuscript.
  • We were not able to measure the degrees of the eye deflection in each subject nor the the 3D characteristics of the nystagmus. We agree that these characteristics are interesting, but we think that this limitation does not invalidate the results of the study. This study was design from a clinical point of view to obtain conclusions that can be directly extrapolated to clinical practice. During clinical practice, the physicians use videoglasses or examine the eye without any device. Therefore, despite of the experimental interest of your suggestion, we prefer to use simple devices to get practical results. However, we have added the model of the videoglasses that were used.
  1. I am not quite understand why the Fisher’s exact test was used for examining “therelationship between gender and the presence of end-point nystagmus” and “the relationship between the first side tested and the presence of nystagmus”. (Line 237 and 268).

These three variables were measured as dichotomous nominal variables. Since the size of the sample is not huge, we think that Fisher’s exact test is an appropriate significance test.

  1. Please describe SAP tests, HEP tests, and SAP to HEP tests in more details. It would be better if these experiments can be illustrated with figures.

Thank you for this suggestion. We have explained in more detail these three tests. We have also included a figure that illustrates each one.

  1. Sampling from staff, patients and patients’ companions in one medical center did not completely fit the healthy population that the study wants to investigate. And there might be a selection bias.

We do not think that we incur in a significant selection bias for having included volunteer subjects from the staff of our hospital and our patients and patient’s companions. All the studied performed in healthy volunteers recruit the subject of their sample somehow. Sampling our whole population looking for healthy volunteers could incur a selection bias too, since polite and collaborative individuals are most likely to participate than those who do not care. We stablished the inclusion and exclusion criteria that assured us that we were including healthy population and we do not think that our recruiting methods had biased the results.

  1. There was a lack of table, figure or literal description to present the demographic features of overall individuals enrolled.

Thank you for this suggestion but we believe that this is not necessary because we described that the population was selected considering gender and age criteria to avoid biases related to the characteristics of our population.

  1. There might be a more detailed description of the stratified sampling technique to confirm it was random.

Thank you for this comment. We do not consider that we had performed a randomized stratified sampling. We simply looked for volunteer subjects that fulfil the inclusion criteria until each of the age and sex groups had six volunteers in it. Since no stratified sampling was done, no changes regarding this issue in the manuscript have been done.

  1. Advantages and drawbacks of the proposed approach should be highlighted.

This is a very interesting comment. We have added a new subheading (4.4) in the discussion of the manuscript that highlights the application of our results in the clinical practice.

Minor problem:

  1. The description of Figure 2 should be more detailed in the manuscript, and whether there is a statistical differences please show it in the figure.
  2. Figure 1, typo error, “above” 80%.
  3. No certain information on location and date was mentioned in the part of materials and methods.

Thank you to recognize these mistakes; we have added the pertinent corrections in the manuscript.

After these corrections, we believe that the comprehension of our manuscript has improved. The authors appreciate the suggestions and ask for new ones if they are considered necessary

Yours sincerily, 

The authors

Reviewer 2 Report

TITLE:

I would suggest you consider revising your title to e.g. “Positional Nystagmus with lateralization of the eyes during positional testing”. At least leave out “maneuvers” and this term is used for treatment and not diagnostics. Also the term “differences from benign paroxysmal positional vertigo” is confusing. If correct, then it should read “differences in positional nystagmus patterns seen with benign paroxysmal positional vertigo

INTRODUCTION:

General issues:

Introduction is quite short. It must be clarified, that in order to place a diagnosis of BPPV, the patient must also have a classical case history compatible with BPPV and not only canal-specific positional nystagmus.

Other studies have dealt with positional nystagmus in healthy adults. That should be mentioned, referred to, and included in the refences. E.g. Martens C et al, Prevalence and Characteristics of Positional Nystagmus in Normal Subjects, Otolaryngol Head Neck Surg. 2016 May;154(5):861-7.

Address possible misinterpretation of positional nystagmus produced if patients, during positional testing, look to the side of the test ear

Classical end-point nystagmus is described and defined with the patient upright! You must write something about this and also acknowledge the fact, that you with this study combine testing of positional nystagmus with lateralization of the eyes.

According to your heading, you also include a comparison with classical BPPV nystagmus. You must then also define it as a secondary endpoint and also include the results on this in the discussion- and conclusion section.

Minor issues:

Line 33: Change ”maneuvers” to ”tests”

Lines 35-37: Add “positional” in front of “nystagmus” (three times in total)

Line 36: Delete ”some”

Line 37: Add ”predefined” in front of “criteria”

Line 45: Within this differential diagnosis” should instead read: “with these differential diagnoses”

Lines 49-51: Move to discussion/conclusion instead

Line 52: Replace “calculate” with “determine”

MATERIALS AND METHODS:

General issues:

vHIT: Several issues must be uniform and should be specified/addressed: same chair used every time?, solid chair?, same lightning? How where the head impulses performed? Same examiner? Experience and education of examiner? Quality of Head impulses? Were head impulses fast, abrupt, unpredictable, low amplitude, high acceleration? Consideration of artifact triggers? Etc.

Lines 92-98:

Consider using a different terminology when you describe what is “normal”. I do not consider glasses during positional testing where fixation is possible/allowed to be normal! Is this set-up trying to mimic a primary, secondary or tertiary test setting? Must be clarified, especially if you want to use the term “normal”!

I do not agree that 40 seconds is the maximum amount of time for nystagmus latency. Please add reference for this or rephrase. I have found a case report that actually describes a latency of five minutes!

Lines 108-110: Does not make any sense. If all Barany Criteria were met, then the patient should have been excluded PRIOR to examination. First diagnostic criteria includes a typical case history of positional vertigo and that should have excluded your patient BEFORE testing. If “only” characteristic positional nystagmus was present, then ALL diagnostic criteria for BPPV would NOT have been fulfilled!

You need to clarify precise what type of goggles that are used and if they magnify the eyes or not

Minor issues:

Lines 59-61: Rephrase whole sentence (not just the correction below). Language is unclear and not correct

Line 60: Replace “showed these nystagmus” with “showed these types of positional nystagmus patterns”.

Line 64: Remove “each”

Line 63-67: Rephrase in order to clarify to the reader, that five groups with different age ranges were included and that each group included six males and six females.

Line 71: should read: “were not included or excluded from the study”

Table 1:

Inclusion criteria need revision. The first and third criteria are agreed criteria – consider removing. Second inclusion criteria should read “Age 20 or above”

Exclusion criteria:

Third exclusion criteria should be corrected: replace “at the time of the test” with “during positional testing”

Fourth exclusion criteria should be corrected: replace ”some” with ”any”

Fifth criteria should read: “Previous ear surgery”

Sixth criteria should read: “Other morphological tympanic membrane alterations besides a monomeric tympanic membrane and/or myringosclerosis”

Seventh criteria should read: Prescription medication of drugs that alter vestibular function (e.g. vestibular suppressants and benzodiazepines)

Eighth criteria should read: Pathological lateral semicircular canal Video Head Impulse Test VOR mean gain values (below 0.80 or above 1.20).

Ninth criteria should read: Fulfilment of BPPV diagnostic criteria

Line 73: Replace “Next” with “Following inclusion”

Line 74: Replace “following” with “according to”

Lines 79-80: Rephrase: “under 80% or over 120%” with “below 0.80 or above 1.20”

Line 104: What does “sense” refer to? Unclear

Line 107: Rephrase: Change “they were suffering from vertigo at that moment” to “experienced any concomitant vertigo”

Line 112: Rephrase. Delete “were” and replace with “was”

RESULTS:

General issues:

Should be shortened a little bit. There is a lot of wording. Only highlight the most important findings - the rest is shown by the figures/tables

Inconsistent and incorrect use of incidence and prevalence

Minor issues:

Line 123: Replace “at that time” with “ at time of inclusion”

Line 124: Replace “on the testing day” with “on the day of testing”

Line 129-130: Rephrase. Change “To describe the nystagmus that were observed, this study uses the eye as a frame of reference” to “With the description of the observed positional nystagmus, the eyes were used as a frame of reference”.

Line 136: Replace “about” to “in”

Lines 132-172: Use “positional nystagmus” consistently instead of just “nystagmus”

TABLE 1:

Table legend should read: “Trial Profile” and nothing else

Use boxes with exclusions on both sides instead of solely on the right side (makes the table easier to read on also makes the table smaller).

Rephrase all exclusion boxes like this (short and precise, numbers centered on a line below the text):

“Sufficient amount of candidates already included (age/gender)”

“n=4”

Line 175: Replace “about” with “in”

Lines 175-210: Use “positional nystagmus” consistently instead of just “nystagmus”

TABLE 2:

Consider revising the heading to be more descriptive of the actual content

Number need to be aligned! Specify in the first column that you include number and percentages like this: E.g. “Torsional clockwise, number (percentage)” and then just put in the numbers in the table. If percentages are includes, do it consistently and also in the same manner (all numbers on one line or two lines)!

Table legend should relate to the table and not additional video coverage which should be placed in the results section instead

Line 219: Replace “were” with “was”

Lines 221-224: Poor English. Rephrase. Also “maneuvers” should be replaced with “positional tests”

TABLE 3:

Heading should be changed, e.g. “Concordance of end-point nystagmus” The rest of the original heading can very well be used as a table legend (that currently is missing)

Commas are not used with decimal numbers in English

Concerning 3.4:

You only included patients with normal mean gain values. Why would you then suspect that any positional nystagmus (in healthy adults) observed would alter the mean gain values or presence of saccades?

“height of the gain” is not the correct terminology. A gain value represents the ratio between eye- and head velocities. How can it be a “height”? Don’t you mean “half the size of the peak head velocity” instead?

DISCUSSION:

General issues:

Comparisons with other studies are missing. Are there no other studies at all on this subject?

Use “positional nystagmus” consistently instead of just “nystagmus”

4.2. Subjective aspects related to the detection of the nystagmus in the end-point position:

You also need to consider that two separate examiners might have different criteria for defining positional nystagmus! How many beats are required? Amplitude? Frequency? Direction? Combination of directions?

Also consider the fact, that when the eye is lateralized, then the possible eye movements will, to a large extent, be purely vertical because this position of the eye minimize any torsional eye movements (this is e.g. also used with 2D vHIT testing in order to minimize noise/artifacts by torsional eye movements)

Videonystagmopraphy can also project the eyes to a big screen that does actually also enable examiners to get a much better view of the eyes – so not only the part on measurements, that is raised, is true here

Discuss further, why there is only found a moderate degree of inter-rater concordance. What could have been done to reach a high degree of inter-rater concordance. Usually many things can be aligned!

A section on “strengths and limitations” with the study is missing – this must be included!

Such a section should also deal with the limitations of your screening that included only questions and one objective examination (vHIT). Did you not perform any other clinical tests? Eye-motility, spontaneous nystagmus, Romberg, Calorics, VEMPs etc? That is definitely also a limitation of the study and might possibly have altered the results!

Minor issues:

Line 289: Change “do” to “does”

Line 291: Change “and the” to “according to”

CONCLUSION:

Line 348: Rephrase “neurotologic”. You have not extensively screened all patients for neither neurological disease or vestibular diseases.

Line 350: Replace “of” with “the” and add an “s” to “characteristics”. Also replace “These nystagmus have“ with “The positional nystagmus observed has…”

Line 351: Replace “show” with “have”

Line 354: Replace “they are asymptomatic” with “patients are asymptomatic” AND also “their” with “nystagmus”

Lines 355 – 357: Video links should be on the same line and not two separate!

Author Response

Dear Dr./Mr./Ms.,

Thank you for giving me the opportunity to submit a revised draft of my manuscript. We appreciate the time and effort that you and the reviewers have dedicated to providing your valuable feedback on the manuscript. We are grateful to the reviewers for their insightful comments on my paper. We have been able to incorporate changes to reflect most of the suggestions provided by the reviewers.

Here is a point-by-point response to your comments and concerns.

  • TITLE: I would suggest you consider revising your title to e.g. “Positional Nystagmus with lateralization of the eyes during positional testing”. At least leave out “maneuvers” and this term is used for treatment and not diagnostics. Also the term “differences from benign paroxysmal positional vertigo” is confusing. If correct, then it should read “differences in positional nystagmus patterns seen with benign paroxysmal positional vertigo

Thank you for your comment. We have changed the title as you have suggested. We think now the title is much more precise.

  • INTRODUCTION: Thank you for all your appreciations.

Introduction is quite short. It must be clarified, that in order to place a diagnosis of BPPV, the patient must also have a classical case history compatible with BPPV and not only canal-specific positional nystagmus.

We have added to our manuscript that it is necessary the presence of a classical history compatible with BPPV to make this diagnosis.

Other studies have dealt with positional nystagmus in healthy adults. That should be mentioned, referred to, and included in the refences. E.g. Martens C et al, Prevalence and Characteristics of Positional Nystagmus in Normal Subjects, Otolaryngol Head Neck Surg. 2016 May;154(5):861-7.Address possible misinterpretation of positional nystagmus produced if patients, during positional testing, look to the side of the test ear. Classical end-point nystagmus is described and defined with the patient upright! You must write something about this and also acknowledge the fact, that you with this study combine testing of positional nystagmus with lateralization of the eyes.

Thank you for your comments. Taking it into account, we have developed this idea on the discussion of our manuscript in section 4.1. Also, we have included a description of the end-point nystagmus in the introduction.

According to your heading, you also include a comparison with classical BPPV nystagmus. You must then also define it as a secondary endpoint and also include the results on this in the discussion- and conclusion section.

As you have recommend we have defined this secondary endpoint. As well we have added a new section in the discussion called 4.4 Apliccation of the results in clinical practice, where we discussed about BPPV and their differences.

All the minor issues where corrected.

  • MATERIALS AND METHODS: We sincerely appreciate all your comments.

vHIT: Several issues must be uniform and should be specified/addressed: same chair used every time?, solid chair?, same lightning? How where the head impulses performed? Same examiner? Experience and education of examiner? Quality of Head impulses? Were head impulses fast, abrupt, unpredictable, low amplitude, high acceleration? Consideration of artifact triggers? Etc.

We have added all the suggested data to our manuscript.

Lines 92-98:

Consider using a different terminology when you describe what is “normal”. I do not consider glasses during positional testing where fixation is possible/allowed to be normal! Is this set-up trying to mimic a primary, secondary or tertiary test setting? Must be clarified, especially if you want to use the term “normal”!I do not agree that 40 seconds is the maximum amount of time for nystagmus latency. Please add reference for this or rephrase. I have found a case report that actually describes a latency of five minutes!

We understand your concerns, thank you for the comments. About the 40 seconds, we now there are some extreme cases, but we established this limit based on the diagnostic criteria of the Barany society where a maximum latency of 40 seconds is determined for the canalolithiasis of the posterior semicircular canal. About the use of the term “normal” we have also modified this lines in the address you have requested.

Lines 108-110: Does not make any sense. If all Barany Criteria were met, then the patient should have been excluded PRIOR to examination. First diagnostic criteria includes a typical case history of positional vertigo and that should have excluded your patient BEFORE testing. If “only” characteristic positional nystagmus was present, then ALL diagnostic criteria for BPPV would NOT have been fulfilled!

We appreciate your suggestion but due to the characteristics of the population on our environment we consider it must be taken into account. Although the presence of BPPV is an exclusion criteria, it is possible that we were faced with a subclinical BPPV that the patient would not have been to express due to a lack of movement or a phenomenon of complacency towards the investigating physicians.

You need to clarify precise what type of goggles that are used and if they magnify the eyes or not

We had clarify the precise type of googles we used in the manuscript.

All the minor issues were corrected. We want to clarify that the term sense. We prefer to use sense instead of direction because the term sense is more specific because it refers to an orientation on a direction. Thus, we believe that this term is more appropriate to describe the vector of the nystagmus.

  • RESULTS: Thanks for your suggestions.

Should be shortened a little bit. There is a lot of wording. Only highlight the most important findings - the rest is shown by the figures/tables

Inconsistent and incorrect use of incidence and prevalence.

We appreciate your recommendations. You are right regarding the use of incidence and prevalence and we have corrected the errors. About the results, we have done small modifications in this sense but not too many, since it would be a conflict with other reviewers.

About Concerning 3.4: This is a study that is part of a larger one, and we have seen that in patients with vestibular neuritis, horizontal positional nystagmus may appear due to this vestibular hypofunction and we believe that they are due to the same pathophysiological mechanism of the one of the head shaking maneuver. This study is being carried out at the same time but it is not finished yet.

The rest of minor issues have been corrected.

  • DISCUSSION: Thank you for all your appreciations.

Comparisons with other studies are missing. Are there no other studies at all on this subject?

As you suggested on the introduction, we have added this idea to our manuscript in section 4.1.

Use “positional nystagmus” consistently instead of just “nystagmus”

This have been checked and corrected.

4.2. Subjective aspects related to the detection of the nystagmus in the end-point position:

You also need to consider that two separate examiners might have different criteria for defining positional nystagmus! How many beats are required? Amplitude? Frequency? Direction? Combination of directions?

It is true that we did not measure the number of beats, the amplitude or the frequency but these test where done by two experienced examiners who measured the direction and duration.

Also consider the fact, that when the eye is lateralized, then the possible eye movements will, to a large extent, be purely vertical because this position of the eye minimize any torsional eye movements (this is e.g. also used with 2D vHIT testing in order to minimize noise/artifacts by torsional eye movements)

We understand that the eye movements can be purely vertical in that position and the torsional ones can be minimize but in this case it is not explained by the results obtained.

Videonystagmopraphy can also project the eyes to a big screen that does actually also enable examiners to get a much better view of the eyes – so not only the part on measurements, that is raised, is true here.

We did used a screen were we could get a better view of the eyes.

Discuss further, why there is only found a moderate degree of inter-rater concordance. What could have been done to reach a high degree of inter-rater concordance. Usually many things can be aligned!

A section on “strengths and limitations” with the study is missing – this must be included!. Such a section should also deal with the limitations of your screening that included only questions and one objective examination (vHIT). Did you not perform any other clinical tests? Eye-motility, spontaneous nystagmus, Romberg, Calorics, VEMPs etc? That is definitely also a limitation of the study and might possibly have altered the results!

 This two points have been taken into account on a new section 4.4 in the discussion were we talk about strengths and limitations.

  • CONCLUSION:

Thank you for your amendments, they have all been taken into account and corrected.

After these corrections, we believe that the comprehension of our manuscript has improved. The authors appreciate the suggestions and ask for new ones if they are considered necessary

Yours sincerily,

The authors.

Reviewer 3 Report

This study describes the caracteristics of the end-point nystagmus during positional testing in healthing individuals who have never suffered from vertigo or other vestibular symptoms and trying to calculate the prevalence, making a very important objective. The references are in general up to date.

However, an increased number of selected ind ividuals might be enrolled with positional maneuvres performed on most of them that have a neurological consultation due to their simplicity and the high chance of diagnosing and treating benign paroxysmal positional vertigo in order to explain the phenomenon as the inner ear could be the main cause of positional end-point nystagmus and not the consequence of the function of the neurons responsible for the tonic intervation,

I finally would like to see a revised form of the manuscript in this direction. 

Author Response

Dear Dr./Mr./Ms.,

Thank you for giving us the opportunity to submit a revised draft of our manuscript. We appreciate the time and effort that you and have dedicated to providing your valuable feedback on the manuscript. We are grateful to the reviewers for their insightful comments on our paper.

After reading your comment, we want to clarify that the calculation of the sample size is a priority in our team. The sample size of 60 individuals was not chosen arbitrarily, but rather was the result of the calculation after setting appropriate confidence intervals.

The sample size is justified as follows. During the planification of the study, we decided that its results would be expressed through percentages with a 95% confidence interval. Our previous and non-systematic observations allowed us to estimate that nystagmus appeared in approximately the half of the subjects. We also estimated that some 12.5% margins of error were useful enough for clinical purposes. Using the binomial distribution to calculate the sample size with 95% confidence, the optimal sample size resulted in 55 subjects. For the sample to be a multiple of 10 (5 age groups · 2 genders), the result was rounded to 60 subjects.

Best regards.

Reviewer 4 Report

i cannot understand the clinical significance of the study.

Author Response

Dear reviewer,   Thank you very much for your dedication to our manuscript. After having read your review and your impressions about it, we have made clarifications about the design of the study and its methodology. In addition, spelling and grammatical errors have been corrected.   In relation to the clinical significance of the article, we want to clarify why we think that the conclusions reached by this study are important. This article describes part of a larger study. This larger study addresses the influence of vestibular function on eye movement control during BPPV testing. The aim of this study is to describe eye movements that can mimic those caused by BPPV. These movements can cause confusion to those less experienced clinicians. Based on the results of this study, we can recommend that when evaluating eye movements in BPPV tests, the eyes must be kept in a neutral gaze position. In extreme gaze positions, a nystagmus may appear. This nystagmus may have some characteristics similar to those of BPPV and thus lead to false diagnoses of BPPV. We have not found any other study that provides recommendations on gaze position during the tests. For this reason, we believe that the study provides a strong experimental basis for making a clinical recommendation.  

Sincerely and thanking you for your time,  

The authors.

Round 2

Author Response

Dear reviewer,

Thank you for reviewing again our manuscript. As suggested, we have checked the spelling and a new version of the manuscript has been submitted.

Sincerely,

The authors.

Reviewer 4 Report

end-point nystagmus is a kind of gaze nystagmus. positional nystagmus is a non-gaze nystagmus.

Author Response

Dear reviewer,

Thank you for reviewing again our manuscript.

The authors have taken into consideration that extreme gaze nystagmus is a gaze nystagmus, and that positional nystagmus is not. Our study is related to the behaviour of this nystagmus while the head is in the Dix-Hallpike position. We have reviewed the introduction and methods of our manuscript and have ensured that this clarification is done.

Sincerely,

The authors.